# Effect of isospin averaging for $ppK^-$ kaonic cluster

**Branislav Vlahovic and Igor Filikhin⋆**

North Carolina Central University, Durham, NC 27707, USA

⋆ ifilikhin@nccu.edu

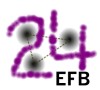
## Abstract

**The kaonic cluster $ppK^-$ is described by isospin-dependent $N\bar{K}$ potentials with significant difference between singlet and triplet components. The quasi-bound state energy of the system is calculated based on the configuration space Faddeev equations within isospin and averaged potential models. The isospin averaging of $N\bar{K}$ potentials is used to simplify the isospin model to isospinless one. We show that three-body bound state energy $E_3$ has a lower bound within the isospin formalism due to relation $|E_3(V_{NN} = 0)| < 2|E_2|$, where $E_2$ is the binding energy of isospin singlet state of the $N\bar{K}$ subsystem. The averaged potential model demonstrates opposite relation between $|E_2|$ and $|E_3(V_{NN} = 0)|$.**

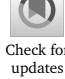

## 1 Introduction

The quasi-bound states in the kaonic cluster $NN\bar{K}(s_{NN} = 0)$ defined by the spin $s_{NN}$ of nucleon pair are intensively debated during the last years. The theoretical predictions for the binding energy are in significant disagreement with the values derived from existing experimental data [1]. The properties of the kaonic cluster are defined by $N\bar{K}$ interaction, having significant difference for the isospin singlet and triplet channels. The isospin singlet component of the $N\bar{K}$ potential generates a quasi-bound state corresponding to the $\Lambda(1405)$ resonance below the $pK^-$ threshold. The resonance has the double state nature due to the $N\bar{K}$ quasi-bound state and $\pi\Sigma$ resonance [2]. There are two potential models for the $N\bar{K}$ quasi-bound state which are used for three-body calculations. The first one is the AY model based on the Akaishi-Yamazaki (AY) $N\bar{K}$ potential with taking into account the $\pi\Sigma$ coupling effectively has been proposed in Ref. [2]. This effective $N\bar{K}$ interactions have a strong attraction in the singlet $I = 0$ channel and a weak attraction in the triplet $I = 1$ channel. The binding energy of $ppK^-$ obtained within this isospin model is $|E_{NN\bar{K}}|$=48 MeV [2]. The two-body threshold is close to the bound state energy of $\Lambda(1405)$ as $K^-p$ bound pair (about 30 MeV). Similar results have been obtained within similar phenomenological models [3,4] taking into account the $\pi\Sigma$ coupling directly. This value is much smaller than the experimentally motivated value of about 100 MeV for the

$ppK^-$ deeply bound state [5–7]. The second model proposed for the $N\bar{K}$ interaction (see HW potential in Ref. [8]) is the chiral model. This model reduces the isospin singlet component of $N\bar{K}$ potential due to the strong coupling $N\bar{K}$ and $\pi\Sigma$ channels. The value about 20 MeV for $|E_{NN\bar{K}}|$ was obtained with the two-body threshold about 11 MeV. Discussion about the experimental background and theoretical interpretations can be found in Ref. [1,9].

In the present work, we show, that within isospin model for $NN\bar{K}(s_{NN}=0)$ based on the phenomenological potentials, the value of binding energy about 100 MeV cannot be reached due to a relation between $E_2$ and $E_3(V_{NN}=0)$. Where, $E_2$ is two-body bound state energy and $E_3(V_{NN}=0)$ is the energy of bound three-body system, when the interaction between the identical particles is omitted. The relation is a result of isospin splitting of $N\bar{K}$ potential and strong binding in the $N\bar{K}$ singlet $I=0$ channel.

The relation between $E_2$ and $E_3(V_{AA}=0)$ has been previously found for bosonic isospinless $AAB$ systems [11]. For the systems, the contribution of the mass polarization term to the three-body energy leads to the relation $|E_3(V_{AA}=0)| > |2E_2|$.

The "isospinless model" for the kaonic clusters based on the isospin averaged $N\bar{K}$ potential have been proposed in Refs. [10, 11]. In Ref. [12], such averaging defined as "V-averaging" is related to isospin averaged $N\bar{K}$ potential: $V_{N\bar{K}}^{av} = \frac{3}{4}V_{N\bar{K}}^{I=0} + \frac{1}{4}V_{N\bar{K}}^{I=1}$. Another type of the averaging called in Ref. [12] as "t-averaging" is applied for two-body $t$-matrix within the impulse representation for treatment of the system. These two types of averaging were proposed for simplification of isospin models describing three and four -body kaonic clusters. The t-averaging was previously used in Refs. [13–15] for $NN\bar{K}$ calculations when $s_{NN}=1$ and $s_{NN}=0$. The $N\bar{K}$ interaction amplitude was presented by isospin decomposition of two-body isospin singlet and triplet amplitudes: $f_{N\bar{K}} = \frac{3}{4}f_{N\bar{K}}^{I=0} + \frac{1}{4}f_{N\bar{K}}^{I=1}$. The decomposition is different for $s_{NN}=0$ and $s_{NN}=1$ spin states of $NN\bar{K}$ system. Based on this difference, the authors of Refs. [13–15] obtain approximate evaluation for strength of the $N\bar{K}$ interaction in the $ppK^-$ and $dK^-$ systems.

We apply the V-averaging to obtain an isospinless model for $NN\bar{K}(s_{NN}=0)$ system. The goal is to compare the isospin and the isospinless model to show the effect of isospin splitting of the $N\bar{K}$ interaction. The result of such comparison is the different relations between $E_2$ and $E_3(V_{NN}=0)$ satisfying for both types of the $N\bar{K}$ potential (AY and sHW). Our study is based on the Faddeev equations in configuration space [16]. The Faddeev equations allow to separate components of the total wave function corresponding to the different particle rearrangements.

## 2 Formalism

### 2.1 Faddeev equation for $AAB$ system

The kaonic cluster $ppK^-$ are represented by the three-body $AAB$ system with two identical particles. The total wave function of the $AAB$ system is decomposed into the sum of the Faddeev components $U$ and $W$ corresponding to the $(AA)B$ and $A(AB)$ types of rearrangements: $\Psi = U + W \pm PW$, where $P$ is the permutation operator for two identical particles. In the expression for $\Psi$, the sign "+" corresponds to two identical bosons, while the sign "−" corresponds to two identical fermions, respectively. Each component is expressed by corresponding Jacobi coordinates. For a three-body system with two identical particles the set of the Faddeev equations is presented by two equations for the components $U$ and $W$ [10, 11, 17]:

$$
\begin{aligned}
(H_0 + V_{AA} - E)U &= -V_{AA}(W \pm PW), \\
(H_0 + V_{AB} - E)W &= -V_{AB}(U \pm PW),
\end{aligned}
\tag{1}
$$

where again the signs "+" and "−" correspond to two identical bosons and fermions, respectively and $H_0$ is the kinetic energy operator presented in the Jacobi coordinates for correspond-

ing rearrangement. The wave function of the system $AAB$ is symmetrized with respect to two identical bosons, while it is antisymmetrized with respect to two identical fermions.

In the presented work, we consider the $s$-wave approach for the $AAB$ systems. For bosonic system, we have the Faddeev equations (1) in the form with the sign "+". When $V_{AA} = 0$, one obtains a single equation: $(H_0 + V_{AB} - E)W = -V_{AB}PW$. Here, we assume that the $V_{AB}$ potential generates a deep bound state with energy $E_2$. The term of right hand side of the equation is the exchange term. This term adds negative energy to the two-body energy $E_2$ defined by left hand side of the equation and the three-body energy becomes less than $E_2$: $E = E_3 < E_2$ (the mass polarization effect). The strength and range parameters of the $AB$ potential and mass ratio $m_B/m_A$ have importance here. The evaluations for the mass polarization term for different systems one can find in Ref. [11].

## 2.2 Isospin formalism for kaonic system

The $NN\bar{K}$ system is a system with two identical particles described by Eq. (1). The separation of spin-isospin variables leads to the following form of the Faddeev equations:

$$
\begin{aligned}
(H_0 + V_{NN} - E)\mathcal{U} &= -V_{NN}D(1+p)\mathcal{W}, \\
(H_0 + V_{N\bar{K}} - E)\mathcal{W} &= -V_{N\bar{K}}(D^T\mathcal{U} + Gp\mathcal{W}),
\end{aligned}
\tag{2}
$$

where $\mathcal{W}$ is a column matrix with the isospin singlet and triplet coordinate dependent parts of the Faddeev component $W$. The component $U$ is presented by isospin triplet part of $\mathcal{U}$ corresponding to the spin singlet state of $NN$ pair. The matrixes have the following form:

$$
D = (-\frac{\sqrt{3}}{2}, -\frac{1}{2}), \quad G = \begin{pmatrix} \frac{1}{2} & \frac{\sqrt{3}}{2} \\ \frac{\sqrt{3}}{2} & -\frac{1}{2} \end{pmatrix}, \quad \mathcal{W} = \begin{pmatrix} \mathcal{W}^s \\ \mathcal{W}^t \end{pmatrix}, \quad \mathcal{U} = \mathcal{U}^t.
\tag{3}
$$

The superscripts $s$ and $t$ in (3) denote the singlet and triplet isospin parts of the components $\mathcal{U}$ and $\mathcal{W}$. In Eq. (2), $V_{NN} = v_{NN}^t$ is isopsin triplet $NN$ potential in the singlet spin state and $V_{N\bar{K}} = diag\{v_{N\bar{K}}^s, v_{N\bar{K}}^t\}$, and the exchange operator $p$ acts on the particles' coordinates only.

For calculations, we use the $s$-wave Akaishi-Yamazaki [2] and the simulating Hyodo-Weise (sHW) effective potentials [18] of $N\bar{K}$ interactions, which are energy independent and include the coupled-channel dynamics into a single channel $N\bar{K}$ interaction. Below, we show that the relation

$$
|E_3(V_{NN} = 0)| < 2|E_2|
\tag{4}
$$

takes place for the kaonic cluster $ppK^-$. Here, it is assumed that the interaction between two identical particles is omitted, $V_{NN} = 0$ and the $|E_3(V_{NN} = 0)|$ is binding energy of the three-body system. The relation (4) can be explained by strong attraction of the isospin singlet $N\bar{K}$ potential having a deep bound state with the binding energy $E_2$.

## 2.3 Reduction to isospinless model: averaged potential

In this section, we define the effective potential obtained by averaging of the initial potential over isospin variables. This averaging produces the "isospinless" (or "bosonic") model for the kaonic clusters.

The isospin averaged potential $V_{\bar{K}N}^{av}$ is defined as:

$$
V_{\bar{K}N}^{av} = \frac{3}{4}v_{N\bar{K}}^s + \frac{1}{4}v_{N\bar{K}}^t.
\tag{5}
$$

Here, we use the isospin singlet and triplet components $v_{N\bar{K}}^s$ and $v_{N\bar{K}}^t$ of the AY $N\bar{K}$ potential. This potential has a moderate attraction in comparison with the strong attraction in the $I = 0$

channel. The two-body threshold is changed to lower one and is not related to the $pK^-$ bound state as $\Lambda(1405)$.

Eq. (2) is reduced to the scalar form by an algebraic transformation. Taking into account that $\widetilde{\mathcal{W}} = D\mathcal{W}$, $V_{N\bar{K}}^{av} = DV_{N\bar{K}}D^T$ and $DD^T = 1$, $DV_{N\bar{K}}GD^T = V_{N\bar{K}}^{av}$ one obtains

$$
\begin{aligned}
(H_0^U + V_{NN} - E)\mathcal{U} &= -V_{NN}(1+p)\widetilde{\mathcal{W}}, \\
(H_0^W + V_{N\bar{K}}^{av} - E)\widetilde{\mathcal{W}} &= -V_{N\bar{K}}^{av}(\mathcal{U} + p\widetilde{\mathcal{W}}).
\end{aligned}
\tag{6}
$$

Thus, the isospin averaging of $N\bar{K}$ potential is defined as $V_{N\bar{K}}^{av} = DV_{N\bar{K}}D^T$.

One can evaluate the mass polarization in the three-body system described by Eq. (6) using the definition: $\Delta = 2E_2^{av} - E_3^{av}(V_{NN} = 0)$. Here, $E_2^{av}$ is $N\bar{K}$ two-body binding energy obtained with the averaged potential and $E_3^{av}(V_{NN} = 0)$ is the three-body binding energy calculated by Eq. (6) when the $NN$ interaction is omitted. The value of $\Delta$ is positive [11].

## 3   Numerical results

The results of the calculations for the $NN\bar{K}$ ground state energy are presented in Table 1. For the both potentials AY and sHW, the relation $2E_2 - E_3(V_{NN} = 0) < 0$ is satisfied. The three-body binding energy $|E_3|$ is larger than the value $|E_3(V_{NN} = 0)|$ due to the contribution of weak attractive $V_{NN}$ potential. Obtained results are comparable with the results of calculations performed within different approaches. For example, calculated values $|E_3|$ reported in Ref. [4] are 47–54 MeV for the phenomenological $\bar{K}N$ potentials that does not exceed the value of 60 MeV.

Table 1: Ground state energies $E_3$ of the $NN\bar{K}$ system with the AY and sHW potentials for the $N\bar{K}$ interactions and the MT I-III potential [19] for the $NN$ interaction. The results for the case $V_{NN} = 0$ are given in parenthesis. The difference $\delta$ of the two-body $2E_2$ and three-body $E_3$ energies, $\delta = 2E_2 - E_3$, is presented. The energies are given in MeV.

| Potentials | $E_2$ | $E_3$ | $\delta$ |
|---|---|---|---|
| MT I-III, AY | -30.30 | -46.0 (-42.9) | -14.6 (-17.6) |
| MT I-III, sHW | -11.16 | -21.0 (-17.1) | -1.3 (-5.2) |

We calculated the difference $\delta$ of the two-body $E_2$ and three-body $E_3$ energies related to Eq. (4) as $\delta = 2E_2 - E_3$ to evaluate the relation (4) for different $N\bar{K}$ potentials. For both models (AY and sHW), the relation is satisfied.

We illustrate the existence of the lower bounds for the ground state energy of the $NN\bar{K}$ system in Fig. 1 and 2 using the AY, sHW and averaged (av) potentials for $N\bar{K}$ interaction. The energies $E_2$, $2E_2$ and $E_3$ are shown as functions of the scaling factor $\alpha$ which controls the strength of interaction between non-identical particles: $V_{N\bar{K}} \to \alpha V_{N\bar{K}}$. The case, when the $AA$ potential acting between identical particles is neglected, $V_{NN} = 0$, is presented in Fig. 1. One can see that the relation (4) is well satisfied for both models with the AY and sHW potentials. The isospinless model with averaged (av) potential demonstrates opposite relation. When $0.9 < \alpha < 1.1$, the mass polarization term depends weakly on strength of the $N\bar{K}$ potential and $2E_2^{av} - E_3^{av}(V_{NN} = 0) \approx const$.

The situation is slightly altered when the $NN$ interaction is included in the calculations as is shown in Fig. 2. The attractive $NN$ interaction affects the $E_3$ and the corresponding curves

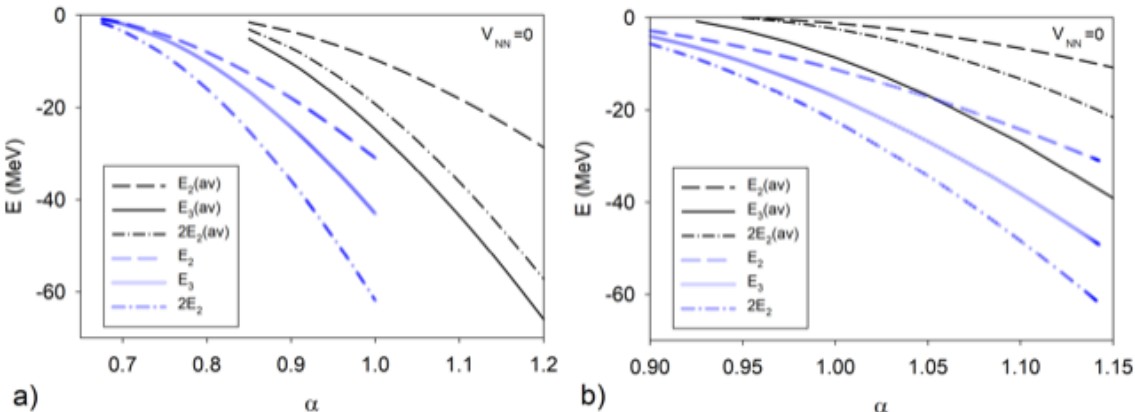

Figure 1: $NN\bar{K}(V_{NN} = 0)$ system: the energies $E_2$ (dashed line), $2E_2$ (dot-dashed line) and $E_3$ (solid line) are shown as functions of the scaling factor $\alpha$, $V_{N\bar{K}} \to \alpha V_{N\bar{K}}$, a) for AY andAY averaged (av) $N\bar{K}$ potentials, b) for sHW and sHW averaged (av) $N\bar{K}$ potentials. The $NN$ potential acting between nucleons is neglected, $V_{NN} = 0$.

become lower comparing with Fig. 1. The relation (4) is well satisfied for the large values of the two-body ground state energy, $|E_2| > 10$ MeV. In the sector of weak $AB$ potential, the opposite relation $|E_3(V_{NN} = 0)| > 2|E_2|$ is satisfied.

Note here that for the model with the averaged (av) potential, the $E_3^{av}$ becomes to closer to $2E_2^{av}$ in the sector of large strength of $N\bar{K}$ potential. It can be explained by the core effect of the $NN$ potential which only appears for the isospinless model. The repulsion of the core plays a role when three-body system is very compact. It is will seen for the strong AY interaction in Fig. 2a).

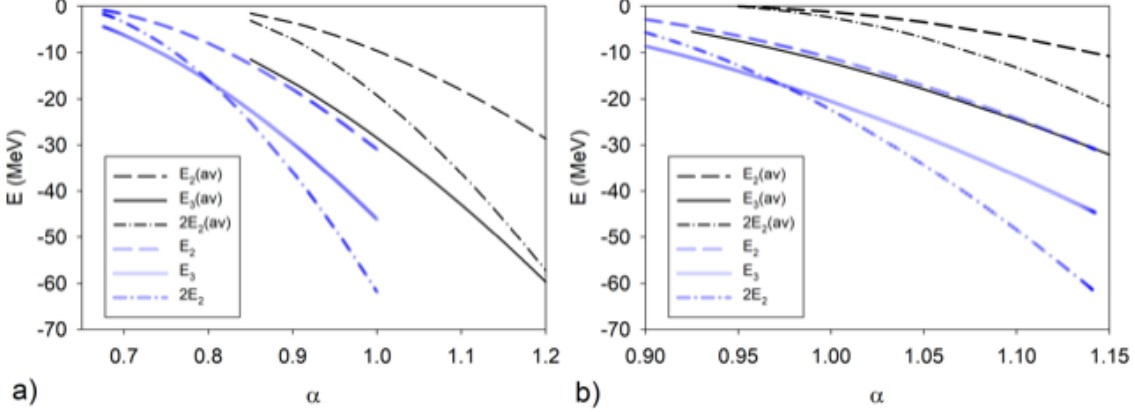

Figure 2: $NN\bar{K}$ system: the energies $E_2$ (dashed line), $2E_2$ (dot-dashed line) and $E_3$ (solid line) are shown as functions of the scaling factor $\alpha$, $V_{N\bar{K}} \to \alpha V_{N\bar{K}}$, a) for AY and AY averaged (av) $\bar{K}N$ potentials, b) for sHW and sHW averaged (av) $\bar{K}N$ potentials.

The energy $E_3^{av}$ of the averaged potential model is larger always to calculated in the isospin model. This could be expected due to higher position of two-body threshold $E_2^{av}$ of the averaged potential model.

## 4 Conclusions

The kaonic cluster $NN\bar{K}(s_{NN} = 0)$ was described within isospin formalism using the Faddeev equations in coordinate space. We have obtained upper bound for the binding energy of quasi-bound state, $|E_3|$, which can be reached by using this phenomenological isospin-dependent potentials. The relation $|E_3(V_{NN} = 0)| < |2E_2|$ takes a place for the kaonic system. In particular, the calculation gives $|E_3(V_{NN} = 0)| \approx 43$ MeV for AY $N\bar{K}$ potential. The $|E_3|$ has to be slightly larger ($|E_3| = 46$ MeV) than $|E_3(V_{NN} = 0)|$, due to the weak attractive contribution of the $NN$ potential. The value of $E_3$ is smaller than $|2E_2| \sim 60$ MeV (the bound) and is significantly smaller than the "experimentally motivated value" about 100 MeV.

We have compared the isospin and averaged models to show the effect of the averaging (or termination of isospin splitting). The energies calculated for the averaged $N\bar{K}$ potential model satisfy the opposite relation: $\left|E_3^{av}(V_{NN} = 0)\right| > 2\left|E_2^{av}\right|$. The averaged potential reduces the two-body threshold $|E_2|$ to a smaller value and three-body binding energy $|E_3^{av}|$ is significantly smaller comparing to one calculated within the isospin model.

## Acknowledgments

This work is supported by the National Science Foundation grant HRD-1345219 and NASA grant NNX09AV07A.

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
