# Peer review of "Effect of isospin averaging for $ppK^-$ kaonic cluster"

_SciPost Physics Proceedings, doi:SciPost Phys. Proc. 3, 040 (2020)_

## Round 1 · Referee Report · Anonymous (Referee 1) · 2020-1-13

Report

The proceedings contribution "Effect of isospin averaging for ppK- kaonic cluster" by Vlahovic and Filikhin presents a series of calculations associated to isospin averaging in 3-body calculations of hadronic systems. In particular, it discusses two different physics issues: mass polarization in 3-body systems (eg whether trimer binding energies are above or below 2 x dimer energies) and the role of isospin degrees of freedom. The contribution is mostly descriptive and based on a series of specific numerical, theoretical results rather than acting as a wide-ranging discussion of previous literature.

While this is fine for a proceedings contribution, I feel the write-up needs to be improved for clarity and readability. I suggest that the authors discuss more clearly why averaged and non-averaged calculations differ so much in their trimer energy predictions in physically motivated terms.

Requested changes

I provide below a series of minor issues that need to be considered before the paper is published as a proceedings contribution in SciPost:

1) Page 1: I assume s_NN refers to the spin of the two-nucleon system? I suggest the authors clarify the meaning of this variable that is used throughout the paper for clarity.

2) Page 1, middle of paragraph 1: "There are two potential models for the N\bar K...". The authors describe in this paragraph only one potential (the AY potential) and then move to discuss "more complicated models [3,4]". Is the second potential of the "two potential models" here the sHW used later on and introduced here simply as Ref 6? I suggest the authors clarify this point.

3) Page 1, paragraph 2, typo: "due to a relations" > "relation"

4) Page 2, paragraph 2, typo: "two types averaging" > "these two types OF averaging". A bit below, "deference" should presumably read "difference".

5) Page 2, paragraph 2: why is the "t-averaging" discussed here? As far as I can tell, this is not used in the calculations presented here?

6) Page 2, paragraph 3: "Faddeev equations" > can the authors provide more details or, ideally, more references that discuss explicitly the numerical techniques in the solution of these equations as they implement them? Accuracy will presumably be necessary for the detailed energy differences discussed here.

7) Please rewrite the final sentence before the start of section 2.2 - there are typos and it reads awkwardly.

8) Page 3, between Eqs. (2) and (3): "isopin" > "isospin". Below Eq. (3), this is misspelled again as "isopsin"

9) sHW is introduced as an acronym in page 3, but has been used before. More importantly, I don't think the sHW potential has been discussed in the introduction either (where AY is in fact mentioned).

10) Page 4, paragraph 1: the variable \Delta is introduced here, but I don't think it is discussed elsewhere in the paper. Why is it introduced in the first place, and what is its order of magnitude?

11) Section 3: "is larger the" > "is larger than the". A bit below, I don't think I understand the meaning of "exited the values"

12) Page 4: "becomes lower than one is in Fig. 1" needs rewritings

  • validity: ok
  • significance: ok
  • originality: ok
  • clarity: low
  • formatting: good
  • grammar: reasonable

Author:  Igor Filikhin  on 2020-02-05  [id 727]

(in reply to Report 1 on 2020-01-13)
Category:
answer to question
correction

Dear Editor Thank you very much for your response. We are grateful to the referee for his/her careful reading of our manuscript and for the valuable feedback we have received, which we have used to make our manuscript more clear and comprehensive. We completely agree with referee’s concerns and we have considered and addressed all comments made by the referee. Below we provide point-by-point responses, a summary of the changes made in the resubmitted manuscript and our response to the referee. Sincerely, B. Vlahovic, I. Filikhin

---

## Round 1 · Referee Report · Anonymous · 2020-1-13

The proceedings contribution "Effect of isospin averaging for ppK- kaonic cluster" by Vlahovic and Filikhin presents a series of calculations associated to isospin averaging in 3-body calculations of hadronic systems. In particular, it discusses two different physics issues: mass polarization in 3-body systems (eg whether trimer binding energies are above or below 2 x dimer energies) and the role of isospin degrees of freedom. The contribution is mostly descriptive and based on a series of specific numerical, theoretical results rather than acting as a wide-ranging discussion of previous literature.

While this is fine for a proceedings contribution, I feel the write-up needs to be improved for clarity and readability. I suggest that the authors discuss more clearly why averaged and non-averaged calculations differ so much in their trimer energy predictions in physically motivated terms. Requested changes

I provide below a series of minor issues that need to be considered before the paper is published as a proceedings contribution in SciPost:

1) Page 1: I assume s_NN refers to the spin of the two-nucleon system? I suggest the authors clarify the meaning of this variable that is used throughout the paper for clarity.

Comment: it is corrected to “The quasi-bound states in the kaonic cluster $NN{\bar K}(s_{NN}=0)$ defined by the spin $s_{NN}$ of nucleon pair”

2) Page 1, middle of paragraph 1: "There are two potential models for the N\bar K...". The authors describe in this paragraph only one potential (the AY potential) and then move to discuss "more complicated models [3,4]". Is the second potential of the "two potential models" here the sHW used later on and introduced here simply as Ref 6? I suggest the authors clarify this point.

Comment: it is corrected to “The first one is the AY model based on the Akaishi-Yamazaki (AY) $N{\bar K}$ potential that takes into account the $\pi\Sigma$ coupling effectively, explained in Ref. \cite{YA07}.” Also, we have corrected the sentence, “Comparable results have been obtained within similar phenomenological models \cite{D17,RS14} taking into account the $\pi\Sigma$ coupling directly.” And we have corrected the sentence, “The second model proposed for the $N{\bar K}$ interaction (see HW potential in Ref.\cite{HW}) is the chiral model. This model” to avoid confusing readers.

3) Page 1, paragraph 2, typo: "due to a relations" > "relation" Comment: it is corrected.

4) Page 2, paragraph 2, typo: "two types averaging" > "these two types OF averaging". A bit below, "deference" should presumably read "difference". Comment: it is corrected.

5) Page 2, paragraph 2: why is the "t-averaging" discussed here? As far as I can tell, this is not used in the calculations presented here?

Comment: it is corrected: “Another type of the averaging noted in Ref. \cite{MAY14} as "t-averaging" is applied for two-body $t$-matrix within an impulse representation for treatment of the system.” to show, that the t-averaging is related to the impulse representation. We used the coordinate representation, and such averaging is not possible.

6) Page 2, paragraph 3: "Faddeev equations" > can the authors provide more details or, ideally, more references that discuss explicitly the numerical techniques in the solution of these equations as they implement them? Accuracy will presumably be necessary for the detailed energy differences discussed here. Comment: Due to the 6-page limit for the presentation we here include only the references in the text: “the set of the Faddeev equations is presented by two equations for the components $U$ and $W$ \cite{K2015,FKSV17,14}:” and added the references in the list of references: \bibitem{FaddeevConfigurSpace} L.D. Faddeev and S.P. Merkuriev, { Quantum Scattering Theory for Several Particle Systems} (Kluwer Academic, Dordrecht, 1993) pp. 398, \doi{10.1007/978-94-017-2832-4}. \bibitem{14} I. Filikhin, A. Gal and V.M. Suslov, Phys. Rev. C {\bf 68} 024002 (2003), doi{10.1103/PhysRevC.68.024002}.

7) Please rewrite the final sentence before the start of section 2.2 - there are typos and it reads awkwardly.

Comment: it is corrected to “The strength, and range parameters of the $AB$ potential and mass ratio $m_B/m_A$ have importance here. The evaluations of the mass polarization term for different systems one can find in Ref. \cite{FKSV17}.”

8) Page 3, between Eqs. (2) and (3): "isopin" > "isospin". Below Eq. (3), this is misspelled again as "isopsin" Comment: it is corrected.

9) sHW is introduced as an acronym in page 3, but has been used before. More importantly, I don't think the sHW potential has been discussed in the introduction either (where AY is in fact mentioned). Comment: it was corrected. We have added the acronym HW in introduction “(please see HW potential in Ref.\cite{HW})” where the HW potential was mentioned. The reference for the potential is included (see in the text: “This energy depended model reduces the isospin singlet component of $N{\bar K}$ potential due to the strong coupling $N{\bar K}$ and $\pi\Sigma$ channels. The value about 20~MeV for $|E_{NN{\bar K}}|$ was obtained with the two-body threshold about 11~MeV. Discussion about the experimental background and theoretical interpretations can be found in Ref. \cite{G2016,GHM}”). The sHW potential is a restriction for the HW potential (see in the text: “the simulating Hyodo-Weise (sHW) effective potentials \cite{JK} of $N\bar{K}$ interactions, which are energy independent and include the coupled-channel dynamics into a single channel $N \bar{K}$ interaction.”). The main property (small binding energy of NK singlet state) of the HW potential is kept by the sHW potentials.

10) Page 4, paragraph 1: the variable \Delta is introduced here, but I don't think it is discussed elsewhere in the paper. Why is it introduced in the first place, and what is its order of magnitude?

Comment: the \delta accompanies the main conclusion of the manuscript proposed for the isospin model and is presented as the relation between E2 and E3. We added the sentence to the text (see “We calculated the difference $\delta$ of the two-body $E_2$ and three-body $E_3$ energies related to Eq. (\ref{eq:11}) as $\delta=2E_{2}-E_{3}$ to evaluate the relation (\ref{eq:11}) for different $N{\bar K}$ potentials. For both models (AY and sHW), the relation is satisfied.”). The magnitude of \delta is defined in the caption to Table 1 and evaluated in the table.

11) Section 3: "is larger the" > "is larger than the". A bit below, I don't think I understand the meaning of "exited the values"

Comment: we corrected to the sentence “As an example, for the phenomenological ${\bar K}N$ potentials calculated values $|E_3|$ do not exceed the value of 60~MeV, as it is reported in Ref. \cite{RS14} they are 47--54~MeV.”

12) Page 4: "becomes lower than one is in Fig. 1" needs rewritings

Comment: we corrected to “The attractive $NN$ interaction affects the $E_3$, and the corresponding curves become lower in comparison with the curves in Fig. \ref{fig 2}.”

Attachment:

answer_to_ref_averaged-potB1.pdf

Author:  Igor Filikhin  on 2020-01-29  [id 722]

(in reply to Report 1 on 2020-01-13)
Category:
answer to question

Dear Editor We are grateful to the referee for his/her careful reading of our manuscript and for the valuable feedback we have received, which we have used to make our manuscript more clear and comprehensive. We completely agree with the referee’s concerns and we have considered and addressed all comments made by the referee. Below we provide point-by-point responses, a summary of the changes made in the resubmitted manuscript and our response to the referee. Sincerely, B. Vlahovic, I. Filikhin

---

## Round 1 · Referee Report · Anonymous · 2020-1-13

The proceedings contribution "Effect of isospin averaging for ppK- kaonic cluster" by Vlahovic and Filikhin presents a series of calculations associated to isospin averaging in 3-body calculations of hadronic systems. In particular, it discusses two different physics issues: mass polarization in 3-body systems (eg whether trimer binding energies are above or below 2 x dimer energies) and the role of isospin degrees of freedom. The contribution is mostly descriptive and based on a series of specific numerical, theoretical results rather than acting as a wide-ranging discussion of previous literature.

While this is fine for a proceedings contribution, I feel the write-up needs to be improved for clarity and readability. I suggest that the authors discuss more clearly why averaged and non-averaged calculations differ so much in their trimer energy predictions in physically motivated terms. Requested changes

I provide below a series of minor issues that need to be considered before the paper is published as a proceedings contribution in SciPost:

1) Page 1: I assume s_NN refers to the spin of the two-nucleon system? I suggest the authors clarify the meaning of this variable that is used throughout the paper for clarity.

Comment: it is corrected to “The quasi-bound states in the kaonic cluster $NN{\bar K}(s_{NN}=0)$ defined by the spin $s_{NN}$ of nucleon pair”

2) Page 1, middle of paragraph 1: "There are two potential models for the N\bar K...". The authors describe in this paragraph only one potential (the AY potential) and then move to discuss "more complicated models [3,4]". Is the second potential of the "two potential models" here the sHW used later on and introduced here simply as Ref 6? I suggest the authors clarify this point.

Comment: it is corrected to “The first one is the AY model based on the Akaishi-Yamazaki (AY) $N{\bar K}$ potential that takes into account the $\pi\Sigma$ coupling effectively, explained in Ref. \cite{YA07}.” Also, we have corrected the sentence, “Comparable results have been obtained within similar phenomenological models \cite{D17,RS14} taking into account the $\pi\Sigma$ coupling directly.” And we have corrected the sentence, “The second model proposed for the $N{\bar K}$ interaction (see HW potential in Ref.\cite{HW}) is the chiral model. This model” to avoid confusing readers.

3) Page 1, paragraph 2, typo: "due to a relations" > "relation" Comment: it is corrected.

4) Page 2, paragraph 2, typo: "two types averaging" > "these two types OF averaging". A bit below, "deference" should presumably read "difference". Comment: it is corrected.

5) Page 2, paragraph 2: why is the "t-averaging" discussed here? As far as I can tell, this is not used in the calculations presented here?

Comment: it is corrected: “Another type of the averaging noted in Ref. \cite{MAY14} as "t-averaging" is applied for two-body $t$-matrix within an impulse representation for treatment of the system.” to show, that the t-averaging is related to the impulse representation. We used the coordinate representation, and such averaging is not possible.

6) Page 2, paragraph 3: "Faddeev equations" > can the authors provide more details or, ideally, more references that discuss explicitly the numerical techniques in the solution of these equations as they implement them? Accuracy will presumably be necessary for the detailed energy differences discussed here. Comment: Due to the 6-page limit for the presentation we here include only the references in the text: “the set of the Faddeev equations is presented by two equations for the components $U$ and $W$ \cite{K2015,FKSV17,14}:” and added the references in the list of references: \bibitem{FaddeevConfigurSpace} L.D. Faddeev and S.P. Merkuriev, { Quantum Scattering Theory for Several Particle Systems} (Kluwer Academic, Dordrecht, 1993) pp. 398, \doi{10.1007/978-94-017-2832-4}. \bibitem{14} I. Filikhin, A. Gal and V.M. Suslov, Phys. Rev. C {\bf 68} 024002 (2003), doi{10.1103/PhysRevC.68.024002}.

7) Please rewrite the final sentence before the start of section 2.2 - there are typos and it reads awkwardly.

Comment: it is corrected to “The strength, and range parameters of the $AB$ potential and mass ratio $m_B/m_A$ have importance here. The evaluations of the mass polarization term for different systems one can find in Ref. \cite{FKSV17}.”

8) Page 3, between Eqs. (2) and (3): "isopin" > "isospin". Below Eq. (3), this is misspelled again as "isopin" Comment: it is corrected.

9) sHW is introduced as an acronym in page 3, but has been used before. More importantly, I don't think the sHW potential has been discussed in the introduction either (where AY is in fact mentioned). Comment: it was corrected. We have added the acronym HW in introduction “(please see HW potential in Ref.\cite{HW})” where the HW potential was mentioned. The reference for the potential is included (see in the text: “This energy depended model reduces the isospin singlet component of $N{\bar K}$ potential due to the strong coupling $N{\bar K}$ and $\pi\Sigma$ channels. The value of about 20~MeV for $|E_{NN{\bar K}}|$ was obtained with the two-body threshold about 11~MeV. Discussion about the experimental background and theoretical interpretations can be found in Ref. \cite{G2016,GHM}”). The sHW potential is a restriction for the HW potential (see in the text: “the simulating Hyodo-Weise (sHW) effective potentials \cite{JK} of $N\bar{K}$ interactions, which are energy independent and include the coupled-channel dynamics into a single channel $N \bar{K}$ interaction.”). The main property (small binding energy of NK singlet state) of the HW potentials is kept by the sHW potentials.

10) Page 4, paragraph 1: the variable \Delta is introduced here, but I don't think it is discussed elsewhere in the paper. Why is it introduced in the first place, and what is its order of magnitude?

Comment: the \delta accompanies the main conclusion of the manuscript proposed for the isospin model and is presented as the relation between E2 and E3. We added the sentence to the text (see “We calculated the difference $\delta$ of the two-body $E_2$ and three-body $E_3$ energies related to Eq. (\ref{eq:11}) as $\delta=2E_{2}-E_{3}$ to evaluate the relation (\ref{eq:11}) for different $N{\bar K}$ potentials. For both models (AY and sHW), the relation is satisfied.”). The magnitude of \delta is defined in the caption to Table 1 and evaluated in the table.

11) Section 3: "is larger the" > "is larger than the". A bit below, I don't think I understand the meaning of "exited the values"

Comment: we corrected to the sentence “As an example, for the phenomenological ${\bar K}N$ potentials calculated values $|E_3|$ do not exceed the value of 60~MeV, as it is reported in Ref. \cite{RS14} they are 47--54~MeV.”

12) Page 4: "becomes lower than one is in Fig. 1" needs rewritings

Comment: we corrected to “The attractive $NN$ interaction affects the $E_3$, and the corresponding curves become lower in comparison with the curves in Fig. \ref{fig 2}.”

Attachment:

SciPost_VF_04.pdf

---

## Editorial Decision

published